# A Fine-Grained Secure Service Provisioning Platform for Hypervisor Systems

**Junho Seo** [1], **Seonah Lee** [2,*], **Ki-Il Kim** [3] **and Kyong Hoon Kim** [4,*]

1 Department of Informatics, Gyeongsang National University, Jinju 52828, Korea; joy2net@gmail.com
2 Department of AI Convergence Engineering (Graduate) and Aerospace and Software Engineering (Undergraduate), Gyeongsang National University, Jinju 52828, Korea
3 Department of Computer Science and Engineering, Chungnam National University, Daejeon 34134, Korea; kikim@cnu.ac.kr
4 School of Computer Science and Engineering, Kyungpook National University, Daegu 41566, Korea
* Correspondence: saleese@gnu.ac.kr (S.L.); kyong.kim@knu.ac.kr (K.H.K.); Tel.: +82-053-950-5554 (K.H.K.)

**Abstract:** As computing technology has been recently widely adopted, most computing devices provide security-related services as basic requirements, which is an important research issue for sustainability of computing devices. The rapid increase of software components makes it difficult to detect or prevent vulnerabilities in the large-size software. One of the prominent approaches for ensuring secure service is the isolation of service which allows the related code and data to be executed only in a particular area. In this paper, we provide a secure service provisioning platform for hypervisor systems. The main contribution of the proposed framework is to enhance the previous secure service provisioning platform in order to solve the non-preemption problem of secure services. Thus, the proposed framework improves the security isolation service in hypervisors and can be used for fine-grained secure service in secure embedded systems.

**Keywords:** trusted service execution; security isolation; hypervisor; fine-grained secure service; secure service provisioning platform

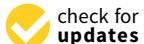



## 1. Introduction

Recently, personal computing devices such as smart phones, tablets, and laptops have been widely used. Most of these devices are always connected to the Internet to provide services such as e-mail, banking, and e-commerce. Although such seamless connection provides convenience to individuals, this causes a severe problem since users' personal information can be taken over by vulnerability attack of service. This risk of vulnerable service takes an important role in sustainability of computing systems. The services use personal information which can be leaked through adversary attacks. Therefore, a considerable amount of research has focused on how to safely execute services to protect personal information [1–6].

In order to run a service safely, the most effective way is to remove any vulnerabilities of the service to be executed. Since adversary attackers usually attack programs using vulnerabilities in the program, creating a program without any vulnerabilities would be the ideal way. However, as the size of software becomes larger and its complexity grows exponentially, eliminating the vulnerabilities requires enormous effort and cost. Thus, recent research has focused on the isolation of secure service from other applications.

One prominent approach for ensuring a secure service execution is to isolate service execution from the normal execution environment. This *security isolation* method allocates service codes or data to a secure area separated from the normal execution environment and blocks abnormal access from non-authorized applications [1,7]. The secure area is accessible only in a determined path and prevents unauthorized access. Using this method, even if the normal execution environment is modified by vulnerability, the secure area

is not affected and thus it can be protected. In addition, since the service using personal information is performed only within the secure area, it is possible to protect the personal information by blocking unintended modification.

Many studies have proposed security platform using this secure isolation technique [2–6,8–10]. One example is the XMHF (eXtensible and Modular Hypervisor Framework) proposed by Vasudevan et al. [4,5]. XMHF uses virtualization technology to achieve isolation by allocating the secure area to the hypervisor separated from virtual machines. The hypervisor is software that provides abstraction of hardware to operating systems and performs resource management, such as redistribution of abstract resources, virtual machine scheduling, and event handling, which are goals of virtualization technology. The guest OS in the virtual machine is not directly accessible to the hypervisor because the hypervisor is implemented in a privileged layer for management. Using this feature, XMHF assigns secure services to the hypervisor area and allows access to the secure services only via a special command called `vmcall` [4,5]. In one recent work in [11], they developed *uberXMHF* architecture for supporting commodity compatibility including Linux and Windows based on XMHF so that they showed the practicality of a secure micro-hypervisor system. In [12], they proposed a mixed-trust computing framework that combines verification and protection for applications with trusted and untrusted execution parts.

In the security isolation approach, however, there exists a critical problem that the normal OS hangs when the normal OS requests the secure service. For example, uberXMHF [11] uses a mechanism called *core quiescing* in order to ensure the isolated execution of a secure service in the running core by stalling other cores. The main reason for this problem is because the normal OS cannot occupy the CPU resources. When the normal OS requests a secure service, the CPU switches to the privilege mode for executing security services in the hypervisor. However, the normal OS cannot be performed in privilege mode. Thus, when the hypervisor is in operation, the normal OS cannot be scheduled and hangs until it receives the service result.

Thus, we provide a modified secure service framework in order to solve the problem mentioned above. The main contributions of this paper include:

- to propose a secure service provisioning platform for solving non-preemptive execution of services,
- to guarantee the execution time of the normal OS while providing the hypervisor-level security services, and
- to implement the proposed framework.

The remainder of this paper is organized as follows: Section 2 provides the background information about security isolation. We define the problem of secure execution framework in Section 3, and propose our secure service provisioning platform for executing fine-grained secure services in Section 4. In Section 5, we show how our framework is implemented. The evaluation of the proposed framework is described in Section 6. Finally, Section 7 discusses several remaining issues such as secure execution on multi-cores and concludes the paper.

## 2. Backgrounds

### 2.1. Virtualization

Virtualization is a technology of abstracting computer resources to hide the physical resources and redistribute them into logical resources. Most of the resources required for the platform to run, such as CPU occupancy, memory space, and storage space, are abstracted. Using this technology, multiple logical platforms can be executed in a single physical system. Logical platforms called virtual machines are executed independently using the allocated resources. The virtual machines have their own OS called guest OS. The resources allocated by a virtual machine are partitioned resources so that it is not possible to access other virtual machines.

The hypervisor is a monitoring program that redistributes abstract resources through virtualization and manages the execution of virtual machines. The hypervisor provides a service in response to resource requests or service execution requests from virtual machines in a form similar to a kernel in an OS. The resources of the system are under the supervision of the hypervisor whose area is inaccessible to virtual machines for administrative and security reasons. Virtual machines need an interface to request resources to the hypervisor because they are not directly accessible to the hypervisor's resources and hardware. The hypercall is the interface for this request, so it is possible to communicate a resource request and service request to the hypervisor only through a hypercall. The hypervisor provides the corresponding services according to the type of hypercall and passes the return value of the service to the OS.

For example, Intel has defined the root mode and the non-root mode to separate the hypervisor and the virtual machines in the virtualization architecture [13–15]. The root mode and the non-root mode are similar to the kernel mode and user mode of the OS. Each mode grants privileges differently, such as restrictions on the commands that can be executed. In the root mode, the hypervisor grants all system resources, commands, and so on. On the other hand, in the non-root mode, the use of resources and commands is restricted from virtual machines. Virtual machines can use a hypercall to request resources that require privilege.

Since the root mode and the non-root mode have different privileges, they need the ability to switch to each other's state. In the x86 platform, the *VM entry* and the *VM exit* operation perform the corresponding function. When a virtual machine requests a privilege using a hypercall, a software trap is generated on the CPU to perform a *VM exit* function. The *VM exit* function switches the state of the CPU to the root mode and calls the handler implemented in the hypervisor. The hypervisor handler parses the requested hypercall and performs the service using the root mode privileges. After switching to the non-root mode in which the system enters the virtual machine, the virtual machine performs the return operation of hypercall with limited privilege. If the virtual machine accesses an area to be executed in the root mode or uses a privilege command, the CPU generates a trap and performs a *VM exit* to notify the hypervisor that an exception handling routine has occurred. The hypervisor performs the appropriate exception handling routine to prevent the virtual machine from accessing the privilege area [13].

Recent processors have been developed with virtualization instructions. Describing virtualization technology as hardware instructions has advantages such as faster execution of the virtualization platform and more extensive resource distribution. In addition, isolating service routines using virtualization instructions can increase reliability by ensuring isolation and performance at the hardware level. The benefits of this virtualization technology are being applied in a variety of areas, and a lot of research is studying the security platform with virtualization. Representative examples include the dynamic analysis of malicious code using virtual machine, event logging analysis through virtualization, and analysis through a system playback function of virtualization [16–19].

*2.2. Security Isolation*

The security isolation technique uses a strategy of isolating the service from the vulnerable OS and avoiding the attack, rather than solving the vulnerability of the OS. Although it is difficult to protect the entire mobile OS, it is more practical to limit the impact of possible vulnerability. In general, OS has a vast amount of code, so it is difficult to verify all the codes in the security aspects. However, the amount of source code of secure service is part of the whole OS. In addition, the security isolation platform is lightweight because it is written in code for security service execution. Therefore, the amount of code to be verified is less than that of the OS source code, so that there is little chance of being exposed to the vulnerability.

In the security isolation platform, the TCB (Trusted Computing Base) is defined as the area where security is completely maintained regardless of whether other domains

are vulnerable to attack if the secure service routine is fully implemented in terms of security and reliability [20–22]. A TCB is a secure area from a security attack because it is isolated from the area normally used by the user. Therefore, secure services, secure libraries, and drivers should be designated as TCB. As the size of the TCB increases, it becomes difficult to secure the reliability. Therefore, the reliability of the entire system is increased by minimizing the TCB size.

Based on the security isolation method, researchers designed various secure execution architectures. Among them, secure execution using virtualization and sandbox-based secure execution architecture are widely used. Virtualization technology is a technology that allows for running multiple virtual machines on one system by abstracting and redistributing resources provided by modern CPUs. Redistribution of resources is managed by monitoring software called a hypervisor. In this architecture, virtual machines or hypervisors have the advantage of being isolated from each other, so that the secure service can be isolated from the environment used by other users. Another architecture is the sandbox-based architecture in which the secure service is executed by using special space provided in CPU design. This space is only accessible with privileged commands, so the secure service can be easily isolated.

### 2.3. Hypervisor Based Secure Execution

The hypervisor-based secure execution platform and secure OS-based secure execution platform use virtualization technology to achieve isolation of secure service. In a virtualization platform, virtual machines do not have direct access to the hypervisor area. This is because the hypervisor executes with special privilege. The hypervisor-based secure execution method uses this feature to achieve isolation by placing secure services in a hypervisor area. Therefore, even if the normal OS is modified by vulnerability, secure service can maintain its reliability because the secure service is not affected.

In [1,23], they address that a lot of research has been proposed for the secure execution method with virtualization technology. Using this architecture, many studies have proposed a hypervisor-based security service platform, which aims to minimize the security threat by reducing the size of the TCB. XMHF [5] is one of the representative lightweight hypervisor-based security frameworks for the research of security services developed by Vasudevan et al. As shown in Figure 1, XMHF achieved being lightweight using the rich single guest OS execution model. The platform was developed to run only a single guest OS in order to simplify guest OS control routines. In addition, the hypervisor delegated control of all the devices in the system to the guest OS, which greatly reduced the complexity by allowing the hypervisor to control only a minimal amount of devices and resources. Thus, the hypervisor achieved being lightweight and reduced the possibility of containing vulnerabilities.

XMHF also uses DRTM (Dynamic Root of Trust for Measurement) technology to provide verification of its integrity at the time of system boot and initialization. DRTM isolates and prevents code from being attacked for reliable execution of code [24], which has been provided by commercial x86 CPUs. In addition, XMHF also uses IOMMU protection technology for input and output memory management. Since XMHF hands over control of the device to the OS, it needs to provide integrity for the memory area.

XMHF provides a module called *hyperapp* that allows for developing and running secure services. The hyperapp supports the security services in the hypervisor area to run in isolation from the guest OS. The guest OS uses a hypercall that is the only communication method to request a hyperapp. When the guest OS invokes the secure service using the hypercall, the XMHF event hub delivers to hyperapp for executing secure service. Secure service invocation and delivery are protected by virtualization, so there is no other way to access secure service in the guest OS. This ensures reliable secure service execution [4,5].

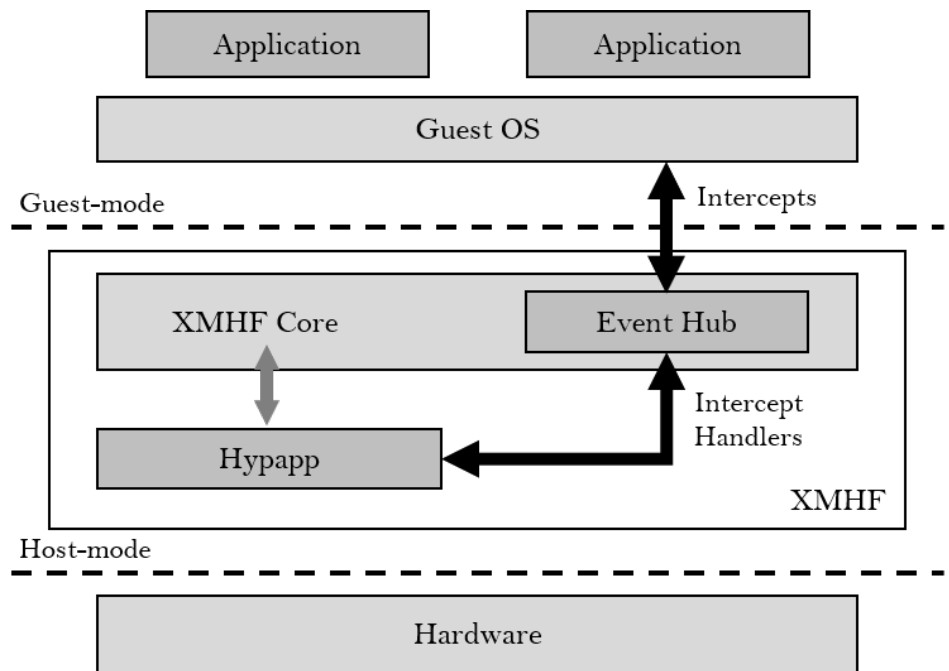

**Figure 1.** An Architecture of XMHF [5].

Another implementation of secure service on hypervisor for trusted mobile computing has been provided in [25,26]. TGVisor is a framework to ensure reliability of a user's geolocation information to provide reliable cloud service in a mobile environment [25,26]. The framework aims to provide cloud service based on geolocation, minimize TCB and verify TCB and geolocation information. To provide cloud service based on trusted geolocation, framework encrypts the user's geolocation data and sends it with geolocation data to the cloud server. The cloud server provides service by verifying the integrity and the possibility of tampering. By reducing TCB, it is possible to reduce vulnerability. In addition, TGVisor performs verification of TCB. It verifies whether the attacker attacks using vulnerabilities of protocols defined by TGVisor and analyzes vulnerabilities. Figure 2 shows the architecture of TGVisor.

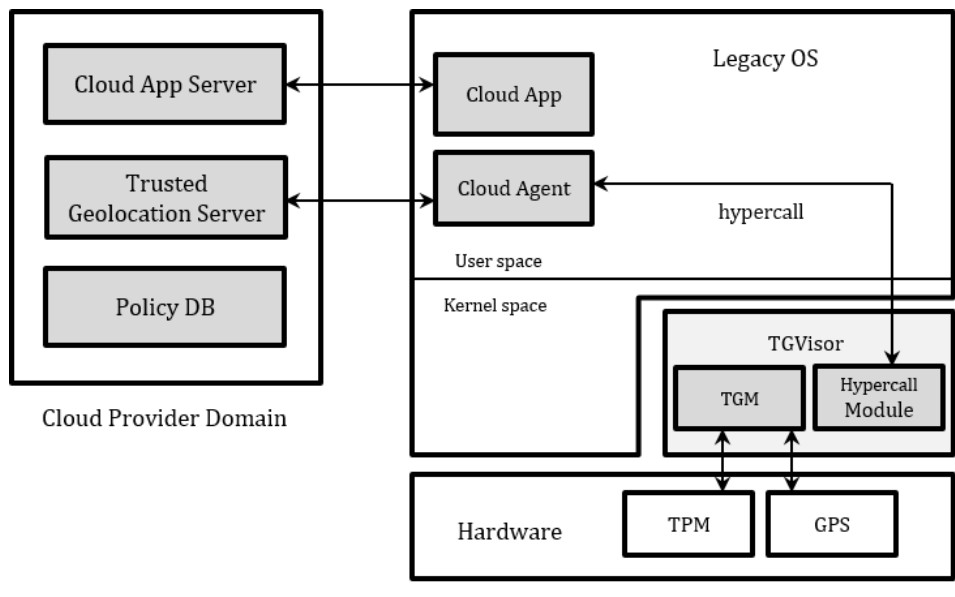

**Figure 2.** An Architecture of TGVisor [25].

As TGVisor framework is implemented using the XMHF framework, it verifies its own code to ensure integrity at boot time. The GPS module is connected to the serial port, and hyperapp can acquire the geolocation information using the serial port. Therefore, it is possible to implement cryptographic service using TPM in hyperapp. As XMHF is a lightweight hypervisor and TGVisor has been developed while being light weight, only a small number of lines of codes are added to reduce the possibility of attack on vulnerabilities.

### 2.4. Secure OS-Based Secure Execution

Another approach for providing secure service is a secure OS-based execution framework. The framework allocates and executes secure service in virtual machines separate from normal OS. In this case, isolation can be achieved because the normal OS does not affect securing OS even if the normal OS is attacked.

A representative platform of secure OS-based secure execution is ARM *TrustZone* [27]. The modern ARM architectures provide for virtualization of a single physical core into two logical cores for the implementation of a secure execution platform. Each logical core is divided into *normal world* for a normal OS execution and *secure world* for a secure OS execution. The two worlds are executed in different privilege modes. The access path between two worlds is managed by software similar to a hypervisor called *secure monitor*. The secure service is performed in secure OS included in secure world. When a secure service is needed, normal OS requests it using a special command called *secure monitor call (SMC)*, which is trapped by the secure monitor for performing the request operation. At the time of the SMC call, the ARM core automatically changes to a privileged mode for secure monitor execution because the secure monitor is included in a secure world. Because of this privilege difference, normal OS cannot be accessed by secure service.

Recently, LTZVisor has been developed based on the TrustZone for providing real-time secure service by porting RTOS to a secure world [2]. The feature of LTZVisor is that secure world has a high priority for CPU preemption to guarantee real-time secure service. The secure OS scheduler is capable of scheduling without being restricted by secure service execution timing using the higher priority. In addition, the normal world where general OS is ported is executed when secure world is in Idle state. Because the secure monitor does not have a scheduler, so the CPU preemption of general OS follows the scheduler policy of RTOS in secure world.

### 3. Problem Definition

As described in the previous section, the secure service in secure execution platform runs in a special space isolated from normal OS. However, the isolation and privilege difference for secure service execution require a certain duration during which the normal OS cannot occupy the CPU. While a secure service is running, the normal OS cannot perform the operation because the OS does not obtain an occupancy of CPU. In particular, on a single-core system, there is only one CPU time to execute one of normal OS and secure service at a certain time. Thus, while a secure service is executed, the normal OS will stop because there is no free time to execute a OS task (see Figure 3). In addition, while the secure service is running, the normal OS has no way to get a privilege from the secure service. Thus, the OS hanging problem persists until the secure service is terminated.

As the execution time of secure service becomes longer, the CPU occupation time of normal OS is limited. If the system only has a single core, the normal OS cannot provide any service for that period. This problem does not affect the communication between normal OS and user if a secure service consumes only a slight time. However, if a secure service occupies the CPU for a long time, users can not do any other work while executing the secure service. In addition, even if the secure service is repeatedly executed, the user experiences inconvenience.

In order to solve this OS hanging problem, we propose a secure service provisioning platform for hypervisor systems. The purpose of proposed framework is to guarantee the execution time of normal OS. The current frameworks do not guarantee normal OS

runtime because they focus on isolation and execution of secure services. Therefore, we limit the execution time of secure service to guarantee the execution time of normal OS within a certain period. In order to limit the execution time, a secure service is divided into a certain size and implemented so that these slices are executed sequentially. Then, when a partitioned service is done, the normal OS is executed. Thus, the execution time of normal OS is guaranteed. On the other hand, since the execution of secure service must be guaranteed, the execution time of normal OS is also limited to be executed for a predetermined time. As a result, the secure service and the normal OS are executed within a period of time.

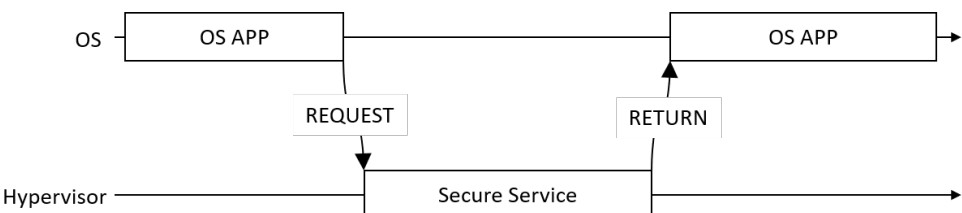

**Figure 3.** A time table of secure service execution—on a normal environment.

## 4. Proposed Framework

In the proposed framework, the secure service is divided into several sub-services in order to limit the preemptive execution time. The normal OS is executed after a slice of secure service is terminated. The normal OS and secure service are executed only for a certain time of period in order to guarantee normal OS and secure service execution, which enables fine-grained service execution. The following sub-sections describe the framework more in detail:

### 4.1. Fine-Grained Service

In the proposed framework, a secure service is divided into several sub-services in order to solve the normal OS hanging problem. In case of a security service with a short execution time of several milliseconds, the CPU is returned to the normal OS immediately, so that it does not interfere with the user. Therefore, the proposed work divides the service routines, which enables the normal OS to preempt the CPU for providing OS service.

A secure service consists of several routines with sequential order. When the all routines are executed in sequence, the service is completed. The service developer develops several routines in a modular form with consideration of this architecture. As shown in Figure 4, developed routines are organized in a chain form and executed in that order. When one routine finishes in one cycle, the next routine is executed in the next cycle. Depending on the implementation of the service, the service routine is implemented to be reusable by changing the order or repeating a routine. When all the routines are finished, the service stores the result and ends. Each service routine is executed at regular intervals. When one service routine is finished, the CPU occupation is passed to the OS. Therefore, the shorter each service routine, the faster the OS can occupy the privilege, which can further reduce the hang of the OS.

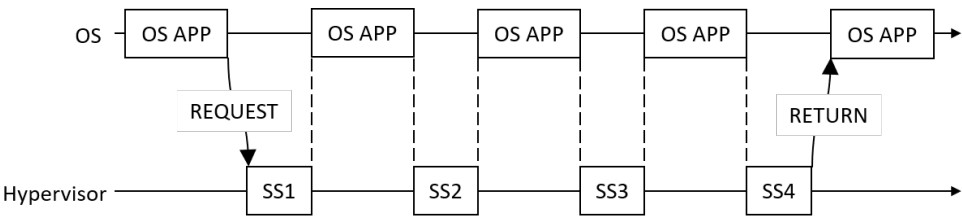

**Figure 4.** A time table of secure service execution—on fine-grained environment.

The partitioned secure service routines are also advantageous for minimizing the vulnerability because each routine is performed independently. In the proposed method,

all service routines are implemented modularly. Routine interactions have no effect other than to save and pass the result. Therefore, since the LOCs for verification are inevitably small, the probability of the vulnerability is reduced.

### 4.2. Secure Service Execution

The previous secure execution platforms perform services on the same core where the application requesting the service is executed. However, the proposed method separates the secure service request and execution for management of fine-grained service execution. When the service manager receives a service request from the normal OS, it returns immediately after the request operation. This allows that the normal OS can use the CPU without wasting time in root mode. While the secure service is running, the application requesting secure service periodically checks the result and receives the result when the secure service is completed.

The service execution is separate from the service request, so it is necessary to store the input value for executing routine. Since the fine-grained secure services are performed independently of each other, the previous routine needs to transfer the data to the next routine. Therefore, we define additional fields in SSCB for storing the values required for service execution. This field contains the value received as an input at the time of service request and the value that is calculated during service execution and passed to the next routine.

### 4.3. Scheduling

Since the execution of normal OS must be guaranteed during the execution of secure service, the secure service completion time is longer than the original secure service completion time. Thus, we made the fine-grained secure services possible to be schedulable, so that they can provide the proper order of secure service according to situation. The scheduling algorithms implemented in the proposed framework are as follows.

#### 4.3.1. First-In First-Out

The secure service inserted first in the service queue is processed first. When a secure service is executed, the other services must wait until the executing secure service is terminated. Since the execution order of the secure service is ensured, the first received secure service is processed quickly, and the result value can be obtained quickly. However, if the execution time of the currently executing service becomes long, the time taken for the service to be executed later is delayed. Therefore, there is a possibility that execution of the entire service is delayed.

#### 4.3.2. Round Robin

All of the fine-grained secure services in the service queue are executed once in order. When a routine of a partitioned service finishes execution, it is re-inserted to the end of the service queue and waits for the next turn without performing the next routine. The algorithm ensures that all the fine-grained secure services run fairly. However, as more services are queued in the service queue, the time required to perform a service becomes longer, so that it takes more time to obtain the result of the entire service.

#### 4.3.3. High-Priority First

The secure services are given priorities and the service having the highest priority is executed first. The services are sorted by the priority in service queue. The service manager takes the highest-priority service from the sorted queue and runs the service first.

## 5. Implementation

We developed the proposed architecture based on the XMHF framework. Figure 5 shows the framework of proposed architecture. The framework is based on hypervisor and runs the existing OS as a virtual machine. The hypervisor consists of *XMHF-core* for a

secure platform, *Service Request Management (SRM)* for managing security service requests and executions, and *Secure Service Pool (SSP)* for implementing secure services.

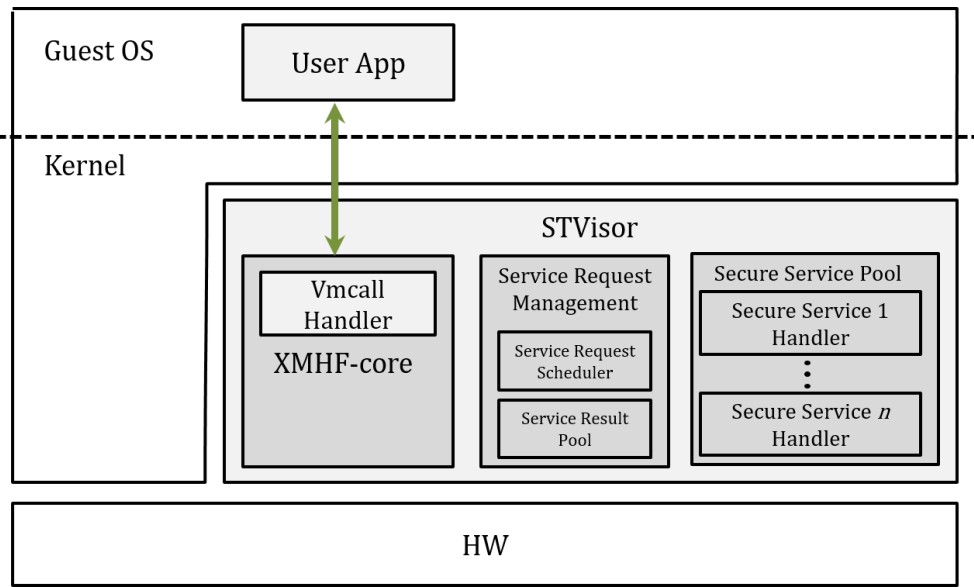

**Figure 5.** The architecture of the proposed framework.

### 5.1. Secure Service

The XMHF provides a service for secure execution called hyperapp. The hyperapp can be loaded in the hypervisor area and executed independently of normal OS. However, XMHF provides only a service as a framework for a single service. Thus, we modified this framework to be able to perform multiple services.

Since the XMHF is designed for providing a single service, there is no need to distinguish between services. Thus, the hyperapp runs immediately upon receipt of a service request via hypercall. In the proposed framework, however, it is necessary to distinguish services when requesting service. For service classification, we assigned each service with a unique number. The service number is entered as a parameter of the service request, and this parameter is used to distinguish which service is required. SSCB is also assigned with a service ID to distinguish service execution. Each running service is identified by the service ID, so that even the same service can be classified as a separate service according to the request.

In addition, we modified hypercall to use for service requests and return results. The normal OS calls the desired service using hypercall, and then periodically calls hypercall to check if the service result is ready to be returned. To distinguish between service call and service result confirmation, we added a parameter to distinguish it. The SRM performs the corresponding work by using this parameter. Figure 6 shows the main components and service flows in the proposed framework.

### 5.2. Service Request Manager

The SRM is a manager that creates and executes a service requested by a user and returns the result value. The framework needs to distinguish services because users can run multiple services simultaneously. Therefore, the proposed method uses the Secure Service Control Block (SSCB) for management. A SSCB is a basic unit for scheduling services, and consists of basic information for providing services. The SSCB is created at the time of service request, which is used to distinguish each service. Because a secure service has a unique SSCB, some secure services performing the same operation are treated as a different service according to the SSCB. Thus, secure services can be executed simultaneously by managing them as service control blocks.

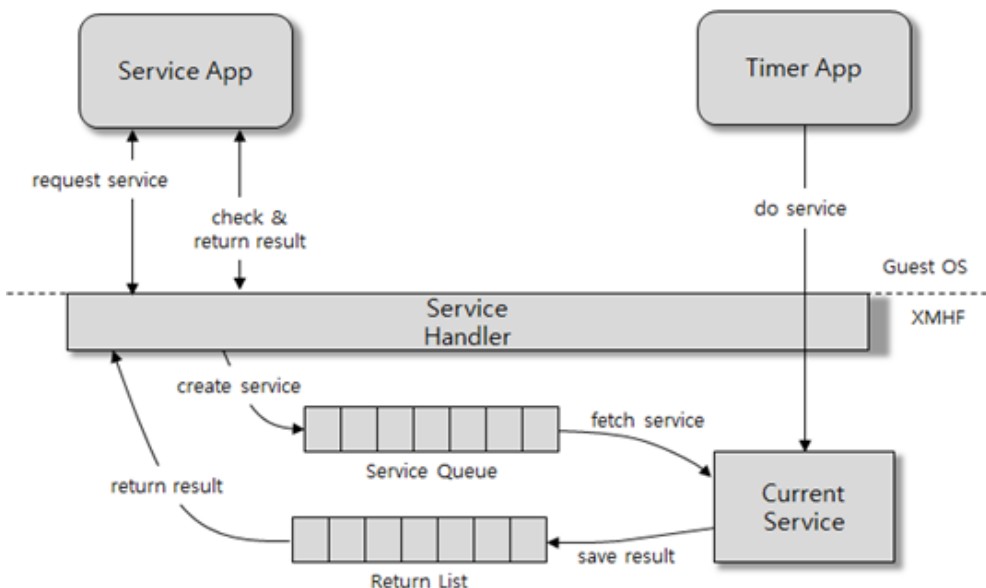

**Figure 6.** Components and service flows in the implementation.

As shown in Figure 7, the generated SSCB is inserted into the queue and waits for execution. The SRM dequeues and executes the SSCB inserted in the service queue according to the scheduling policy. In order to simplify the service execution routine, we defined a single queue for inserting a service. This is to avoid increasing the vulnerability due to increased LOC when queue management becomes complicated. In addition, we defined a result list to store the execution result of the service. The application calling secure service inquires the execution status of service periodically to check whether the service is completed. The service manager retrieves the result of corresponding service from the result list and returns the result value.

In the proposed method, fine-grained services should be executed periodically. The services are implemented in the hypervisor, but the OS occupies most of the time on the system. Thus, the framework needs a trigger to request the execute of service periodically. The trigger uses the timer interrupt to periodically invoke service execution regardless of normal OS execution. However, the hypervisor has limited use of the timer due to the limitations of XMHF architecture. Therefore, the implemented framework replaces trigger by implementing periodic call using vmcall instead of timers.

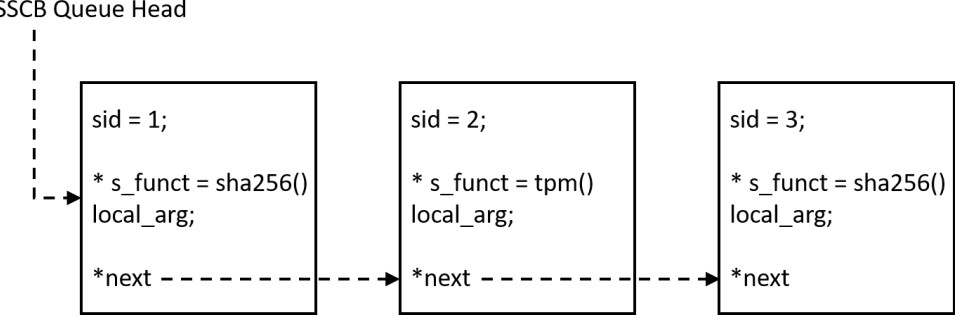

**Figure 7.** Secure service queue with SSCB.

*5.3. Secure Service Pool*

The SSP is an area where a service developer implements a secure service. This area stores service codes for executing a service when a user requests. A partitioned service has several routines whose addresses are stored and managed in the SSCB. The SRM reads the

information of current service execution recorded in the SSCB and calls the corresponding routine using a stored address.

In the SSP, a service consists of an initialization routine, a terminate routine, and intermediate routines. The initialization routine is called when the SSCB is created. This routine stores input data received from the normal OS and initial values needed for execution of secure service routines. Intermediate routines are routines that need to be computed for service provision. The developer develops routines that divides a service into appropriate function units. The developer can specify the next routine to execute multiple routines in order or the current routine recursively at the end of the intermediate routine. Finally, the terminate routine is invoked when the operation of secure service is completed and performs the task to save the result data in result list. After this routine is executed, the SSCB is deleted and the service execution is terminated.

## 6. Evaluation

In order to evaluate that the secure service provisioning platform is working properly, we implemented the SHA256 encryption algorithm as a secure service. The SHA256 encryption algorithm is an algorithm that encrypts original data with abbreviated data using a hash function. The abbreviated data are a unique value. If the original data are changed even by one bit, the abbreviated data become a totally different value. This characteristic is used to verify the authenticity of the original data. We implemented this algorithm by separating it into five routines. The first routine is a routine for initialization and is performed at the time of service request. The last routine is a routine for returning results. Thus, the actual routine for executing the service consists of three routines.

Based on the implemented framework and secure services, we measured code length and execution time. We show that the size of the TCB is small through code length measurement. In addition, we analyzed the service execution overhead by measuring execution time.

### 6.1. Minimized TCB

Since the proposed framework is a framework for secure execution, the length of source code affects TCB reliability. We have measured the source code length of the implemented framework using *cloc* command in linux to prove that the size of TCB is small.

Table 1 shows the lines of source codes of the proposed framework. The implemented framework consists of the existing XMHF core part, the secure service handler that implements the proposed method, and the secure service routines that implement the developed service. The XMHF core part is a hypervisor part for secure execution service. For reliability, we have not modified this part. We only implemented the proposed method in hyperapp, which is the handler part that executes the developed service routines. Therefore, the proposed method with a scheduler is implemented in the hyperapp handler part. The service function is the part where the developed service routines of the service to be provided is implemented, and the sha256 function we have implemented for testing is written.

**Table 1.** Lines of code in the implemented region.

| Part | Lines of Code |
|:---:|:---:|
| XMHF core | 6018 |
| hyperapp handler | 198 |
| service function | 330 |

As shown in Table 1, the handler part added to implement the proposed method is about 3.2% of the whole framework and has a very small number of lines. This shows that the implementation of the proposed method is consistent with the tendency to reduce the TCB for reliability.

### 6.2. Performance

We measured the execution time of the implemented service to check the performance of the proposed framework. One to four services were executed simultaneously, and the OS guaranteed time of the framework was set to 1 s. That is, the routines of service are executed periodically at intervals of 1 s. The service to be executed simultaneously called the same SHA256 service.

Table 2 shows the response times of first and last services for the number of services. As the number of services increases, the amount of time that all services are terminated increases in proportion to the number of services. Although the services run at the same time in Table 2, the response times are different according to the scheduling policy, FIFO (First-In First-Out) or RR (Round-Robin). For example, when four concurrent services are executed, the first service finishes at about 3 s in FIFO policy. On the contrary, the first one in RR policy ends at about 9 s. Let us note that we do not include the execution results of the priority-based scheduling scheme because the results are similar to FIFO policy except the order of services.

**Table 2.** Response times with a single timer.

| Number of Services | | 1 | 2 | 3 | 4 |
|---|---|---|---|---|---|
| FIFO | First service | 3000.72 | 3000.7 | 3000.76 | 3000.9 |
| | Last service | - | 6001.2 | 9002.85 | 12,002.25 |
| RR | First service | 3000.77 | 5001.04 | 7001.37 | 9001.73 |
| | Last service | - | 6001.15 | 9001.77 | 12,002.21 |

(time : ms).

As we measured the scenarios, we also observed the hanging phenomenon of OS. As a result, the OS is delayed slightly at intervals of 1 s because the implemented framework is set up to perform service execution every one second. Thus, the OS is guaranteed to occupy the CPU except the requesting time. Although the service response time is more than 3 s, as shown in Table 2, the normal OS does not hang up for the whole response time but is delayed just for the service execution time every one second.

The service time or OS hanging time consists of execution time of secure service routine, hypercall time to execute secure service, and time for handling secure service. As shown in Table 2, a maximum of 2850 us is required excluding the time required to guarantee OS execution.

We also experimented with increasing the number of execution times on which the service is running. Table 3 shows the results of execution time in which service runs two times per period based on the experiment in Table 2.

**Table 3.** Response times with double timers.

| Number of Services | | 1 | 2 | 3 | 4 |
|---|---|---|---|---|---|
| FIFO | First service | 1500.5 | 1500.49 | 1500.51 | 1500.5 |
| | Last service | - | 3000.75 | 4501.01 | 6002.25 |
| RR | First service | 1500.5 | 2500.72 | 3500.92 | 4501 |
| | Last service | - | 3000.74 | 4501.11 | 6001.42 |

(time : ms).

As a result of measurement, the response time of service was half of the previous execution time. In addition, the execution overhead excluding OS execution time is not very different from the previous experiment. During the experiments, we found that there is no severe OS hanging problem at the double rate execution of the services.

## 7. Discussion and Conclusions

### 7.1. Discussion

In this paper, the proposed method is based on a single core platform where the core can only perform one task at a time. Thus, if the core executes a secure service, the normal OS will stop because there is no extra core to execute. However, current commercial off-the-shell CPUs are mostly multicore. In the multicore, even if one secure service is executed, the normal OS does not stop because the OS can continuously perform operations on other cores. Thus, it seems that the multicore platform is the solution to the OS hanging problem.

However, it is possible to execute multiple services on a multi-core platform. There are many applications running on the normal OS, and many apps require secure services. There are also many kinds of services provided by the secure platform. Thus, if the number of requested services is equal to the number of cores at the same time, the OS may be stopped again. For this reason, the multicore approach is not a fundamental solution.

We expect that, if the fine-grained secure service is extending, it also can support multicore-based security service framework. The proposed framework separates service request and execution, and it is possible to execute appropriately by applying service scheduling. Therefore, it is possible to avoid a case where multiple services are simultaneously executed through scheduling. In addition, it is possible to distribute service execution cores in a balanced manner, thereby preventing degradation of normal OS due to service execution.

In addition, in order to solve the problem that normal OS is hanged due to simultaneous execution of secure services, it is possible to fundamentally solve the problem by limiting the number of concurrent service executing cores. Limiting the number of cores guarantees the normal OS execution because at least one core can execute the normal OS. Alternatively, there is a way to restrict a particular core to execute only secure services. In this case, the performance of the normal OS is deteriorated. However, it is impossible to attack the vulnerability due to shared data in core such as vulnerability attack technique which tracks back the cache at same core, so reliability of secure service is improved.

As a limitation of this study, there is the data race problem of shared resources. Since the proposed method executes on a single core, the problems such as deadlock caused by shared resource usage violation are not serious. However, this issue is expected to become more prominent on a multi-core platform. When a number of services are executed, a race for preemption of a shared device or a resource frequently occurs, which has the possibility to lead to a sharing resource violation. In addition, since the secure service can invade another memory, the memory access restriction should also be considered. Thus, we need to find solutions to these problems.

In the proposed framework, the real-time secure service execution is guaranteed by periodically executing the divided functions which are developed by the service provider. If one of the divided functions runs for a long time, the task in the OS runs out of deadline time, violating the deadline. To solve this problem, we need to extend the framework to execute non-divided secure services for a certain amount of time periodically like scheduler in normal OS. In future studies, we plan to apply this concept to the framework so that real-time is guaranteed even if the developer does not divide the function.

### 7.2. Conclusions

The proposed platform describes how to solve the hanging problem of normal OS on a secure execution platform. The secure execution platform, like XMHF, has a problem that the OS is hanged while the secure service is running. This problem is caused by the difference between execution area and privilege. Due to this difference, the normal OS has limited the way of occupying CPU while secure service is running.

To solve this problem, we propose a fine-grained secure service provisioning platform for hypervisor systems. The proposed platform divides and executes the secure service. In addition, when one partitioned service is terminated, the normal OS is executed to guarantee execution time. Since there is no way for normal OS to occupy CPU itself when

a secure service is executed, we have restricted the execution of secure service to guarantee normal OS execution time.

We implemented the proposed method based on XMHF framework. The implemented framework does not immediately execute the secure service request but stores it in a queue for a waiting of execution. In addition, the scheduler is called periodically to execute the secure services stored in the queue according to scheduling policy. When a developed secure service is executed, the CPU occupation is returned to normal OS, thereby ensuring the execution of the OS.

We measured the execution time of implemented framework and confirmed that normal OS execution time is guaranteed. As an experiment result, although the secure service execution time was long, it was confirmed that the OS execution time is guaranteed during that time. Since we need further investigation on the system overhead, we are working on more experiments to analyze the system performance.

In future studies, we will extend the proposed method to be applied on multicore platforms. In discussion, we described the extensions that we need to apply on multicore, and we need to validate and apply them in future studies. In order to analyze the proposed architecture, we will conduct more experiments for performance evaluation. We also need to discuss the validation of security service implementation for providing vulnerability of secure service software.

Another direction of further research is security analysis of the proposed framework. For example, when real-time tasks with trusted and untrusted execution parts are given to the framework [12], a new scheduling analysis is required to guarantee both real-time and security requirements of tasks.

**Author Contributions:** J.S. and K.H.K. proposed the secure service provisioning platform. J.S. implemented the model, conducted the experiments, and wrote manuscript under the supervision of K.H.K. S.L. assisted and performed model comparison experiments. K.-I.K. reviewed the paper and enhanced the writing. All authors have read and agreed to the published version of the manuscript.

**Funding:** This research was supported by the "Regional Innovation Strategy (RIS)" through the National Research Foundation of Korea (NRF) funded by the Ministry of Education (MOE) (2021RIS-003), and by the National Research Foundation of Korea (NRF) grant funded by the Ministry of Education (NRF-2021R1A2C1094167).

**Conflicts of Interest:** The authors declare no conflict of interest.

## Abbreviations

The following abbreviations are used in this manuscript:

| | |
|---|---|
| VM | Virtual Machine |
| IOMMU | Input/Output Memory Management Unit |
| FIFO | First In First Out |
| RR | Round Robin |
| DRTM | Dynamic Root of Trust for Management |
| TCB | Trusted Computing Base |
| TPM | Trusted Platform Module |
| SMC | Secure Monitor Call |
| SSCB | Secure Service Control Block |
| SRM | Service Request Management |
| SSP | Secure Service Pool |

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
