# Peer review of "A Fine-Grained Secure Service Provisioning Platform for Hypervisor Systems"

_electronics, doi:10.3390/electronics11101606_

Round 1
Reviewer 1 Report
Reviewer Report
Examination of the manuscript shows that the present investigation is good in the present form. but, the following major points should be addressed carefully during the revision to change my decision in the revision:
- Add a nomenclature table with SI units and all used abbreviations.
- Respect the guidelines of the journal and its citation style.
- Provide a suitable reference for each used equation or model.
- The main findings should be highlighted in the abstract.
- The main objectives of the study should be itemized at the end of the introduction.
- The authors have invited to incorporate real images for the realized experiences.
- Correlate the main graphical results by an accurate relationship.
- Improve the discussion more.
- Link the title with the abstract and conclusions.
- Remove all typos and grammatical errors.
- Based on your results, how can the investigators increase the quality of these performance parameters?.
- the paper language should be revised carefully.
Author Response
The authors are very grateful to the reviewer for the helpful comments to enhance the quality of the paper.
Point 1: Add a nomenclature table with SI units and all used abbreviations.
Response 1: As the reviewer suggested, we added the nomenclature table in Table 4 for the terminologies used in the paper.
Point 2: Respect the guidelines of the journal and its citation style.
Response 2: As the reviewer pointed out, we modified all the reference styles as the guidelines of the journal.
Point 3: Provide a suitable reference for each used equation or model.
Response 3: We added the citations of Fig. 1 and Fig. 2.
Point 4: The main findings should be highlighted in the abstract.
Response 4: As the reviewer suggested, we clarified the main contribution of the paper in the abstract.
Point 5: The main objectives of the study should be itemized at the end of the introduction.
Response 5: We clearly described the main objectives of the paper at the end of Section 1, as the reviewer suggested.
Point 6: The authors have invited to incorporate real images for the realized experiences.
Response 6: Since our implementation is the program codes, the real images can be the screen capture of the codes. Thus, we only introduced the figures about the system architecture (from Fig. 2 to Fig. 5) and partly some codes as in Fig. 6.
Point 7: Correlate the main graphical results by an accurate relationship.
Response 7: In this paper, we only used the table for presenting the experimental results. As the results are enough to use tables, rather than graphs, we just leave the table results.
Point 8: Improve the discussion more.
Response 8: We modified the discussion subsection in Section 7.1. We also added more discussion..
Point 9: Link the title with the abstract and conclusions.
Response 9: As the reviewer suggested, we slightly modified the abstract and conclusions.
Point 10: Remove all typos and grammatical errors.
Response 10: We proofread the paper and tried to correct typos.
Point 11: Based on your results, how can the investigators increase the quality of these performance parameters?.
Response 11: The current work just provides the feasibility of the proposed architecture. We will further investigate the performance bottleneck and improve it in future work, which is added in Section 7.2.
Point 12: the paper language should be revised carefully.
Response 12: We proofread the paper and tried to improve the language.

Reviewer 2 Report
- Abstract and conclusions can be explained more.
- Words are repeated even in the same sentences. Try to avoid those repetitions.
Ex: a) The rapid increase of software components in mostly all devices makes it difficult to detect or prevent vulnerabilities in these components.
Ex: ) One of the prominent approaches for ensuring secure service is the isolation of the secure service from the other services.
These repetitions are not only in the abstract, it is found through the manuscript. Try to avoid those repetitions.
- Problem definition can be much clearer.
- The sentence formulation is more formal. It can be improved. For example, First line from the proposed methodology “The proposed framework has the following strategies”. It is more formal.
- Try to include the computational cost for the proposed and existing methodologies.
6. Results and discussion could be more rigorous.
Author Response
The authors are very grateful to the reviewer for the helpful comments to enhance the quality of the paper.
Point 1: Abstract and conclusions can be explained more.
Response 1: As the reviewer suggested, we modified abstract and conclusions more.
Point 2: Words are repeated even in the same sentences. Try to avoid those repetitions.
Ex: a) The rapid increase of software components in mostly all devices makes it difficult to detect or prevent vulnerabilities in these components.
Ex: ) One of the prominent approaches for ensuring secure service is the isolation of the secure service from the other services.
These repetitions are not only in the abstract, it is found through the manuscript. Try to avoid those repetitions.
Response 2: We proofread the paper and tried to correct some errors.
Point 3: Problem definition can be much clearer.
Response 3: As the reviewer suggested, we clarified the contribution of the paper at the end of Section 1.
Point 4: The sentence formulation is more formal. It can be improved. For example, First line from the proposed methodology “The proposed framework has the following strategies”. It is more formal.
Response 4: As the reviewer pointed out, we modified the paragraphs.
Point 5: Try to include the computational cost for the proposed and existing methodologies.
Response 5: The current work just provides the feasibility of the proposed architecture. We will further investigate the performance bottleneck and improve it in future work, which is added in Section 7.2.
Point 6: Results and discussion could be more rigorous.
Response 6: As the reviewer suggested, we modified the results and discussion more.

Round 2
Reviewer 1 Report
The paper can be accepted after minor revisions.
recent references should be added to support the work especially from 2021 and 2022 year.
the paper language should be improved to be suitable for publication in high impacted journal.
more results can be added in details to support the work
Author Response
The authors are very grateful to the reviewer for the helpful comments to enhance the quality of the paper.
Point 1: Recent references should be added to support the work especially from 2021 and 2022 year.
Response 1: As the reviewer suggested, we added more recent references of [8], [9], and [10]. .
Point 2: The paper language should be improved to be suitable for publication in high impacted journal.
Response 2: As the reviewer suggested, we tried to proofread the paper more and enhanced it.
Point 3: More results can be added in details to support the work.
Response 3: Since we need more time for planning and conduction more experiments to analyze the system overhead, we will let it as the future work, which is added in the conclusions.

Reviewer 2 Report
Your effort to improve the manuscript is commendable. I proposed acceptance.
The paper can now be accepted for publication.
Author Response
The authors are very grateful to the reviewer for the helpful comments to enhance the quality of the paper.
As the reviewer suggested, we tried to proofread the paper more and enhanced it.